# Regular Oral Health Management Improved Oral Function of Outpatients with Oral Hypofunction in Dental Hospital: A Longitudinal Study

**DOI:** 10.3390/ijerph19042154

**Published:** 2022-02-14

**Authors:** Yukiko Hatanaka, Junichi Furuya, Yuji Sato, Risako Taue, Yoshiki Uchida, Toshiharu Shichita, Tokiko Osawa

**Affiliations:** Department of Geriatric Dentistry, Showa University School of Dentistry, 2-1-1 Kitasenzoku, Ohta-ku, Tokyo 145-8515, Japan; y.hatanaka@dent.showa-u.ac.jp (Y.H.); sato-@dent.showa-u.ac.jp (Y.S.); gd21-r015@grad.showa-u.ac.jp (R.T.); y.uchida@dent.showa-u.ac.jp (Y.U.); shichita@dent.showa-u.ac.jp (T.S.); tokiko@dent.showa-u.ac.jp (T.O.)

**Keywords:** oral function, oral health, oral frailty, oral hypofunction, oral diadochokinesis, tongue, oral health management, oral health instruction, older people

## Abstract

This longitudinal study aimed to clarify the impact of regular oral health management for oral hypofunction on the oral function of older dental outpatients. The 68 participants enrolled in this study were older dental outpatients (mean age 78.5 ± 8.1 years). According to the number of declined oral examinations after the first exam, participants were assigned to the oral hypofunction group (Hypo group, ≥3), receiving regular oral health management with a leaflet at the dental clinic, or the pre-oral hypofunction group (Pre-hypo group, ≤2), which served as a control. At the second oral examination, after approximately 6 months to 1 year, the Hypo group showed significant improvement in the tongue-lip motor function (Oral diadochokinesis, ODK) /pa/, /ta/, and masticatory function, while the Pre-hypo group showed significant worsening in oral hygiene and oral wetness. Temporal changes in ODK /pa/, /ta/, and the number of declined examination items were significantly different between the groups. Multiple analysis revealed that the number of improved oral examination items were associated with presence of regular oral health management after adjusting for age, sex, number of visits, measuring period, and dental treatment. Regular comprehensive oral health management for oral hypofunction improves and maintains oral function among older dental outpatients.

## 1. Introduction

Oral function is known to decline with aging and systemic disease in addition to oral disease; in particular, masticatory and swallowing dysfunctions due to organic and functional changes of the oral cavity, represented by tooth loss and loss of tongue function, lead to nutritional intake methods and malnutrition [1,2,3,4]. Furthermore, poor oral function is a predictor of physical frailty, sarcopenia, need for nursing care, and mortality [5,6,7]. Accordingly, maintaining healthy oral function is extremely important from the perspective of general health in older individuals. Japan is a forerunner in terms of super-aging societies. The importance of oral frailty [5,6,7], a progressive decline in oral function, ranging from trivial decline to dysphagia, and its countermeasures have been proposed [7].

The management of oral hypofunction in dental outpatient clinics is one of the most important strategies for preventing oral frailty that dentists can implement into routine practice [8,9]. In Japan, oral hypofunction was first defined as a condition of complexly reduced oral functions, which satisfied three or more of the following seven oral examination criteria: oral hygiene, oral wetness, occlusal force, tongue-lip motor function, tongue pressure, masticatory function, and swallowing function. In Japan, oral hypofunction has been included in the national medical insurance as a dental disease since 2018, and the need for oral health management of oral hypofunction in dental care is increasing. The prevalence of oral hypofunction increases with age [8,10].

In oral health management for oral hypofunction, it is important to motivate the patients to engage in oral function training on a daily basis to improve or maintain the oral function of their own accord, similar to tooth brushing in periodontal disease management. Moreover, decreased oral function is related to physical, psychological, and social frailty [7,11,12]. Therefore, it is important to provide comprehensive oral health instruction tailored to individual oral hypofunction, systemic conditions, and lifestyle factors, such as diet and sociality. It has been recommended that oral hypofunction should be regularly managed via oral function examinations approximately every 6 months to monitor the changes in oral function. However, there is no scientific evidence regarding oral health management for oral hypofunction in dental care, and there are only a few reports on the improving effect of oral function instructions [9,13]. Shirobe et al. [13] found that an oral frailty prevention program (comprehensive oral function training) at the dental office improved the oral function of community-dwelling older people. We also clarified the effect of oral health instruction on oral diadochokinesis in our previous study [9], indicating that decreased tongue-lip motor function in older dental outpatients could be improved by oral health instruction.

However, the effectiveness of regular and comprehensive oral health management for oral hypofunction, including the influence of dental treatment and the presence of systemic diseases on oral function, remains unknown. Dental treatment such as prosthetic treatment could improve the criteria for oral hypofunction, such as the occlusal force and masticatory function. In addition, systemic diseases could affect almost all the criteria for oral hypofunction. Although these factors would affect the regular oral health management of oral hypofunction in daily dental care, previous research has not considered them in detail. Therefore, this longitudinal study aimed to investigate the impact of regular comprehensive oral health management for oral hypofunction in older dental outpatients in daily dental care.

## 2. Materials and Methods

### 2.1. Participants

The study participants comprised 105 patients who visited the outpatient clinic of geriatric dentistry at the Showa University Dental Hospital for periodontal or denture maintenance from July 2018 to December 2019 and underwent oral examinations for oral hypofunction [7] for the first time. Excluding 37 patients who met the exclusion criteria, 68 patients (mean age 78.5 ± 8.1 years, 31 men and 37 women) who underwent the second examination after approximately 6 months to 1 year were enrolled in this longitudinal study. The exclusion criteria were patients with incomplete data and patients with severe dysphagia or cognitive decline which made performing the examination difficult. Systemic and oral records were collected from the medical records. Informed consent was obtained from all study subjects using the opt-out method. This study was approved by the International Review Board of the Showa University Dental Hospital. (Approval Number: DH2018-032).

The characteristics of the participants of this study are shown in Table 1. Based on the results of the first seven-item oral examination, the participants in this study were classified into two groups: the oral hypofunction group (Hypo group), which met three or more of the seven items of the diagnostic criteria for oral hypofunction, and the pre-oral hypofunction group (Pre-hypo group), which met the criteria for two or fewer items. The hypo group received regular oral health instructions for oral hypofunction in addition to general cleaning instructions at each visit. This oral health instruction was based on the leaflet published by the Japanese Society of Gerodontology, “For those who were diagnosed with oral hypofunction” [14], and was conducted in the form of oral and written explanations of its contents, including oral hypofunction and lifestyle factors, such as diet and sociability. The time required for instruction depended on the character of the patient; however, the average time was approximately 10 min. Conversely, for the pre-hypo group, the instruction with a leaflet was given only for the items that showed a decline among the seven criteria at the time of the first examination, and only general oral cleaning instructions were given at subsequent visits. In both groups, a second examination was conducted approximately 6 months to 1 year later. During this period, necessary dental treatment was provided when oral problems, such as caries, acute attacks of periodontal disease, and denture fracture, occurred. 

### 2.2. Outcomes

In this study, the data of the seven items of oral examination for oral hypofunction [7], namely, oral hygiene, oral wetness, occlusal force, tongue-lip motor function (oral diadochokinesis (ODK)), tongue pressure, masticatory function, swallowing function, and the presence of hypofunction in each item were collected from medical records. Additionally, the diagnosis of oral hypofunction was collected. The examination and management of oral hypofunction were performed by five skilled dentists based on the diagnostic criteria of Minakuchi et al. [7] using the following examination methods. Oral hypofunction was diagnosed when the total number of items with decreased function was three or more. 

Oral hygiene was assessed using the tongue coating index (TCI), a 3-point scale (score 0, 1, or 2) that evaluates the degree of tongue coating in each area by visual examination of the tongue surface divided into nine sections [15]. Oral uncleanliness, indicating poor oral hygiene, was defined as a TCI of ≥50%. When oral uncleanliness was observed, patients were instructed on oral and denture cleaning habits, tongue coating removal, use of interdental brushes and floss, and gargling habits.

Oral wetness was assessed by measuring the degree of wetness at the center of the dorsum of the tongue thrice using an oral moisture meter (Mucus, Life, Saitama, Japan), and the median value was evaluated as the degree of oral wetness [16]. Oral dryness was defined as a value of <27.0. When oral dryness was observed, patients were instructed to perform appropriate oral exercise habits, water intake habits, salivary gland massage, and moisturizer use.

The occlusal force was measured using a pressure-sensitive sheet (Dental Prescale 2, GC, Tokyo, Japan) and an analyzer (Bite Force Analyzer, GC, Tokyo, Japan) to determine the maximum occlusal force during 3 s of clenching in the intercuspal position [17]. A maximum occlusal force of less than 500 N is considered a decreased occlusal force. For denture wearers, the measurements were obtained with the denture in place. In cases where decreased occlusal force was observed, guidance was given to restore the occlusion through dental treatment, avoiding a crunchy diet, and masticatory muscle training such as chewing.

Tongue-lip motor function was assessed by ODK. The number of syllables pronounced per second was calculated by making the participants pronounce as many single syllables of /pa//ta//ka/ as possible for five seconds and measuring them using an automatic measuring device (Kenkou-kun Handy, Takei Kiki Kogyo, Niigata, Japan) [18]. Decreased tongue-lip motor function was considered to be present if any one syllable of /pa/, /ta/, or /ka/ was pronounced less than six times per second. Training for articulation, rapid speech, lip-and-check muscle, and a range of movements of the tongue and lip was performed when a decreased tongue-lip motor function was observed.

Tongue pressure was measured as the maximum pressure generated when the balloon of the tongue pressure probe was pressed between the tongue and palate for approximately 7 s using a tongue pressure measuring instrument (TPM-01, JMS, Hiroshima, Japan) [19]. Decreased tongue pressure was defined as a maximum tongue pressure of <30 kPa. The participants were instructed to perform tongue resistance exercises when decreased tongue pressure was observed.

The masticatory function was measured as the amount of glucose eluted during the chewing of a gummy jelly. After chewing 2 g of gummy jelly (Glucolum, G.C.) for 20 s, the gummy and water were rinsed with 10 mL of water, and the amount of glucose eluted from the solution passing through the mesh was measured using a chewing ability examination system (Gluco Sensor GS-II; GC, Tokyo, Japan) [20]. A glucose concentration of less than 100 mg/dL indicates a decrease in masticatory function. When decreased masticatory function was observed, the patient was instructed on the recovery of masticatory function through dental treatment, masticatory function training using gum, and food intake habits such as chewing well and maintaining appropriate eating patterns.

Swallowing function was assessed using the subjective swallowing screening questionnaire (The 10-item Eating Assessment Tool, EAT-10) [21]. EAT-10 comprises 10 questions, which can be answered on a 5-point scale (0 = no problem, 4 = severe problem) to evaluate the swallowing function. The maximum score is 40, with a higher score indicating a decrease in swallowing function. Decreased swallowing function was defined as a total score of ≥3. In cases where decreased swallowing function was observed, the patient was instructed to undergo a thorough examination by a dysphagia specialist, swallowing-related muscle exercises, and respiratory exercises.

In addition to oral examinations, the following data were collected: age, sex, comorbidities, smoking status, activities of daily living (ADL), the period between the first and second examinations, number of visits, details of dental treatment, and number of teeth present. Comorbidities were scored using the Charlson comorbidity index (CCI), and the presence of hypertension, diabetes, mental diseases (other than dementia), progressive neurological diseases (Parkinson’s disease, etc.), and oral cancer were also investigated. Hypertension, mental diseases, and progressive neurological disease were considered to be present if they were diagnosed by a physician and medication was allowed. ADLs were classified into three categories: no care required for daily living (0), partial care required (1), and full care required (2). Dental treatment was categorized into three levels: no dental treatment other than maintenance (0), minor dental treatment without occlusal changes, such as resin filling (1), and major dental treatment with occlusal changes, including prosthetic dental treatment, such as denture repair (2). The number of teeth present was defined as the number of teeth excluding those exhibiting vertical movement due to periodontal disease and the remaining roots [7].

### 2.3. Statistical Analysis

The Mann–Whitney U, Chi-square test, and Fisher’s exact test were used for between-group comparisons. Wilcoxon’s signed-rank test was used for within-group comparison of the first and second examinations. Interaction effects were analyzed using a two-factor repeated measures ANOVA to determine the difference in changes over time in both groups. Multiple regression analysis was used to determine the factors associated with improvements in the number of hypofunctional items of seven oral examinations. Statistical analysis was performed using SPSS ver.27 (IBM Japan, Tokyo, Japan), and the significance level for all statistical processing was set at 5%.

## 3. Results

### 3.1. Participants Characteristics

Table 1 shows the characteristics of the study participants. Among the 68 participants in this study, 42 patients were in the Hypo group (mean age 78.2 ± 6.9 years, 17 males and 25 females) met at least three of the diagnostic criteria for oral hypofunction, and 26 patients in the Pre-hypo group (mean age 78.6 ± 8.8 years, 14 males and 12 females) met two or fewer criteria. There were no significant differences at baseline between the two groups, except for the number of teeth present. The median measurement period between the first and second examinations was seven in the Hypo group and eight in the Pre-hypo group. The median number of visits was three in both groups. The mean number of teeth present was 13.2 ± 9.2 in the Hypo group compared to 18.1 ± 8.2 in the Pre-hypo group, indicating that the number of teeth was significantly lower in the Hypo group (*p* = 0.030). Moreover, none of the patients had progressive neurological disorders or oral cancer, and none of them smoked or were dependent.

### 3.2. Comparison of the First and Second Times in the Hypo and Pre-Hypo Groups

Table 2 and Table 3 show the results of the comparison between the first and second examination values in the Hypo and Pre-hypo groups, respectively. In the Hypo group, there was a significant improvement in ODK /pa/ from a median of 5.8 to 6.2 (*p* = 0.001), ODK /ta/ from 5.8 to 6.0 (*p* = 0.017), and masticatory function from 107 to 123 (*p* = 0.028). However, in the Pre-hypo group, oral hygiene worsened from 22.2% to 27.8% (*p* = 0.043), and oral wetness significantly decreased from 30.1 to 28.0 (*p* = 0.004). Table 4 shows that the temporal changes in ODK /pa/ (F = 11.438, *p* = 0.001), /ta/ (F = 6.291, *p* = 0.015), and the number of hypofunctional items (F = 8.671, *p* = 0.004) in both groups showed a significant interaction effect, which means that the trend of temporal changes in each group differed significantly.

Table 5 shows the prevalence of oral hypofunction during the first and second examinations. In the Hypo group, 74% of patients exhibited oral hypofunction at the second examination, while 26% had improved oral hypofunction. Conversely, in the Pre-hypo group, 46% were newly diagnosed with oral hypofunction at the second examination, while 54% remained. 

### 3.3. Factors Associated with Improvements in the Number of Hypofunctional Items over Time

Table 6 shows the results of the multiple regression analysis with improvements in the number of hypofunctional items on oral examination, which showed significant differences in the temporal changes, as the objective variables, and age, sex, CCI, measurement period, number of visits, dental treatment, and presence of regular oral health management for oral hypofunction (0 = Pre-hypo group, 1 = Hypo group) as the explanatory variables. Even after adjusting for these variables, only oral health management for oral hypofunction was significantly associated with improvements in the number of hypofunctional items on oral examination. Furthermore, the number of teeth present, presence of hypertension, presence of diabetes, and dental treatment were not significantly associated with any outcomes related to oral hypofunction in multiple analyses.

## 4. Discussion

To manage oral hypofunction, it is necessary that patients accept and actively work on improving their declining oral function. It is extremely important to maintain the patients’ awareness and motivation for oral function. The results of this study revealed that several oral functions can be improved or maintained by providing regular comprehensive oral health instructions using a leaflet in patients with oral hypofunction in a dental outpatient clinic. The oral health instruction would help motivate patients to improve their oral function in their daily life. In particular, tongue-lip motor function, for which patients can easily perform daily training, was found to improve even within a short period of time. In addition, it could be considered that the improvement in tongue-lip motor function led to improvement in the masticatory function. Conversely, patients who did not receive regular oral health instruction were more likely to have a decline in oral function over time. This tendency was particularly pronounced in the oral environment, including oral hygiene and wetness. Oral function can be measured using the seven criteria for oral hypofunction; however, oral function itself is a comprehensive function. The seven oral examination items are interrelated; therefore, it is important to provide comprehensive oral health instruction for oral function. The importance of oral health instruction for oral function in dental outpatient clinics as well as conventional dental health guidance for oral hygiene was clarified. Oral hypofunction, which is considered a condition in which oral frailty has progressed, may contribute to the prevention of physical and social frailty through the diversity of food intake and malnutrition. Therefore, regular oral health management for oral hypofunction in dental outpatient clinics will be very important in the future super-aging society, and this study is the first to clarify its effect. 

Regarding the improvement in oral hypofunction, specialized dental care may be effective in dealing with individual oral function decline. For example, prosthetic treatment is effective in decreasing the occlusal force and masticatory ability [22,23,24]. Therefore, it is essential to provide appropriate dental treatment to patients with oral hypofunction. Nevertheless, oral hypofunction is a complex condition in which various aspects of oral functions are reduced; although it has clear diagnostic criteria and is considered a reversible dental disease in which early detection and appropriate measures can restore the normal limits of function, unlike dysphagia caused by progressive diseases [7]. Therefore, motivating patients to maintain and improve oral function through self-training and dental treatment, as well as providing comprehensive oral health instructions including lifestyle guidance, and nutritional counseling, are recommended as methods to manage oral hypofunction [25]. Previous reviews have reported that one-on-one dietary counseling in conjunction with dental treatment has the potential to change the patients’ eating habits [26]. In addition, previous studies have shown that dietary and lifestyle guidance, in addition to dental treatment, can improve nutrient intake [27], nutritional status [28], and body composition [29], including weight. Therefore, providing guidance on lifestyle and diet in dental care is effective. In this study, oral health instruction was provided in the form of an explanation regarding the contents of the leaflet “For those who were diagnosed with oral hypofunction” published by the Japanese Society of Geriatric Dentistry. This leaflet contains information on general conditions and lifestyle habits, including diet, as well as oral hypofunction that patients can use to maintain and improve their oral functions, and is written in plain language that is easy for patients to understand. Therefore, this leaflet helped to provide oral health instructions for oral hypofunction in this study.

The participants in this study were independent patients who attended outpatient dental clinics for dental maintenance. Although they were relatively old and had few diseases other than hypertension and diabetes, they were considered robust older individuals. Therefore, it was relatively easy to obtain motivation, which likely remained high in the Hypo group regardless of the number of visits or the measurement period between the first and second examinations. ODK /pa/, /ta/, and masticatory function significantly improved over time in the Pre-hypo group. Considering that mastication is a coordinated movement of the teeth, mandible, tongue, and cheek, there was no significant improvement in the occlusal force. Since the presence of prosthetic dental treatment was not related to the results, it is possible that the improvement in masticatory function in the Hypo group was a result of improved tongue movement function through oral health instruction [9]. A previous study by Shirobe et al. [13] provided oral function training to dental outpatients and reported that the tongue-lip motor function was easily improved by training, which supports the results of this study. Furthermore, in this study, the interaction effect analysis revealed that the improvement in the tongue-lip motor function in the Hypo group differed from that in the Pre-hypo group. It has been reported that range-of-motion training, exercise training, and muscle strength training are effective for tongue-lip motor function, but the instruction for tongue-lip motor function in the leaflet used in this study was relatively easy to implement, such as “Increase opportunities to talk with family and friends”, and this may be one reason why differences in tongue-lip motor function were observed over time in this study [30]. However, as xerostomia is easily influenced by medications and systemic factors [31,32], tongue pressure and occlusal force are affected by systemic muscle strength [33], and swallowing function requires specialized treatment [34,35], it is possible that the simple guidance provided in this study was not enough to improve these items of oral hypofunction. 

None of the items in the Pre-hypo group showed improvement, and oral hygiene and wetness decreased significantly over time, but these were thought to be due to decreased oral health literacy. Oral wetness may be greatly influenced by medication; however, since there was no significant difference in the history of oral dryness between both groups in this study, the worsening of oral wetness in the Pre-hypo group can be attributed to the absence of oral health management. In the Hypo group, oral health instruction may have improved literacy, leading to the maintenance of oral hygiene and oral wetness. Since oral wetness measures the surface moisture of the tongue, the worsening of oral dryness in the Pre-hypo group may be attributed to the worsening of tongue coating adhesion due to oral hygiene, or vice versa. Previous studies have shown that oral dryness worsens tongue coating [36], suggesting that oral hygiene and wetness in oral hypofunction were related. 

Oral hypofunction was diagnosed when three or more of the seven items in each examination were hypofunctional. Therefore, the number of items that fall under the declined category in each examination is important, as it may show the severity of oral hypofunction. In this study, the number of hypofunctional items changed differently over time in both groups, and multiple analysis revealed that the number of hypofunctional items tended to decrease in the Hypo group, suggesting that regular oral health management for oral hypofunction might be effective for older dental outpatients. Approximately a quarter of the Hypo group recovered from oral hypofunction. In contrast, the number of hypofunctional items was significantly increased in the Pre-hypo group, and half of the Pre-hypo group was newly diagnosed with oral hypofunction at the second examination. Therefore, it is suggested that the absence of oral health management could lead to a decline in oral function due to a decrease in the motivation of older outpatients. Our previous study [9] revealed the importance of reconsidering the cut-off value of the seven items and the criterion for three out of seven items necessary for diagnosis. In addition, our previous study revealed several oral examination items for oral hypofunction indicated a relationship between age and sex. Although it is still debatable whether the cut-off values of seven items were validated for diagnosis, seven oral examination items were needed to understand how their oral function declined and how to improve them individually. Thus, the present study suggested the necessity of regular oral health management for older patients, including those without an oral hypofunction diagnosis as well as those with oral hypofunction.

This study had several limitations. As a general rule, individual oral health management was provided to the Hypo group during the visit to the dental hospital, and general oral hygiene instructions were provided to the Pre-hypo group. Since these guidelines were provided when patients visited the hospital, the number of instruction sessions was not constant for all study participants, and the period of the first and second examinations was also not constant. Further, the interventions for the study participants were provided with necessary dental care. Although multiple analysis adjusting for these factors revealed the importance of oral health management, it is necessary to consider future interventional studies that examine the effects of oral health management on oral hypofunction under defined oral conditions among the study participants. Additionally, we did not evaluate the change in the frequency of conversation opportunities and other related factors which were mentioned in the leaflet used in this study and would affect the result of this study. Furthermore, the changes in oral function over a longer period remain unclear. As the oral environment and functions tend to decline in older individuals who require nursing care and hospitalized patients [37,38], it is necessary to clarify the oral health management of these populations in the future.

## 5. Conclusions

In this longitudinal study of dental outpatients, oral hypofunction due to a complex decline in oral function was improved or maintained by regular oral health management for oral hypofunction. Tongue-lip motor function and masticatory function were more likely to improve, whereas oral hygiene and wetness were more likely to worsen without oral health management for oral hypofunction. These findings suggest that it is important to provide regular comprehensive oral health management measures and increase the oral health literacy of older patients by conducting oral function examinations from a stage of normal oral function.

## Figures and Tables

**Table 1 ijerph-19-02154-t001:** Participants Characteristics.

	Hypo Group (N = 42)		Pre-Hypo Group (N = 26)		
	Median (Q1–Q3)	Mean (SD)	Median (Q1–Q3)	Mean (SD)	*p*-Value
Age	80.0	78.6	78.5	78.2	0.672
	(72.8–84.3)	(8.8)	(73.8–83.0)	(6.9)	
Measurement period between the 1st and 2nd examinations (month)	7	7.6	8	8.5	0.135
	(6–9)	(2.1)	(6–10)	(2.2)	
Number of visits	3	5.1	3	4.7	0.338
	(1–8)	(5.1)	(1–8)	(5.5)	
CCI	0	0.6	0	0.4	0.509
	(0–1)	(0.9)	(0–1)	(0.7)	
Number of teeth present	16	13.2	19	18.1	0.030 *
	(3–22)	(9.2)	(12–26)	(8.2)	
	N	%	N	%	
Sex					0.324
Men	17	(40.5)	14	(53.8)	
Women	25	(59.5)	12	(46.2)	
Disease					
Hypertension	19	(45.2)	16	(61.5)	0.220
Diabetes	7	(16.7)	4	(15.4)	1.000
Mental disease	0	(0.0)	2	(7.7)	0.143
Dental treatment					0.857
Maintenance only	23	(54.8)	16	(61.5)	
Minor change	6	(14.3)	3	(11.5)	
Major change	13	(30.9)	7	(27.0)	

* *p* < 0.05; Mann–Whitney U test or Chi-square test, and Fisher’s exact test (hypo group vs. pre-hypo group). Hypo group, patients with oral hypofunction (number of hypofunctional oral examination items ≥ 3); Pre-hypo group, patients without oral hypofunction (number of hypofunctional oral examination items ≤ 2); CCI, Charlson comorbidity index; Q, quartile; SD, standard deviation.

**Table 2 ijerph-19-02154-t002:** Temporal changes in oral function between the 1st and 2nd examination in the Hypo group.

Hypo Group (N = 42)	1st Examination		2nd Examination		
Seven Oral Examination Items for Oral Hypofunction	Median (Q1–Q3)	Mean (SD)	Median (Q1–Q3)	Mean (SD)	*p*-Value
(1) Oral hygiene	27.8	32.3	27.8	34.3	0.443
	(11.1–50.0)	(25.9)	(16.7–52.8)	(23.8)	
(2) Oral wetness	28.1	27.5	27.0	26.9	0.100
	(25.4–30.1)	(4.0)	(25.5–28.7)	(2.5)	
(3) Occlusal force	327.0	408.0	422.5	488.1	0.055
	(214.8–585.3)	(281.1)	(251.5–672.3)	(286.5)	
(4) Tongue-lip motor function					
ODK /pa/ *	5.8	5.5	6.2	6.1	0.001 *
	(5.0–6.3)	(1.1)	(5.6–6.5)	(0.8)	
ODK /ta/ *	5.8	5.5	6.0	5.9	0.017 *
	(4.8–6.2)	(1.1)	(5.6–6.4)	(0.8)	
ODK /ka/	5.3	5.2	5.4	5.4	0.052
	(4.4–5.8)	(1.0)	(5.0–5.9)	(0.8)	
(5) Tongue pressure	23.2	23.9	24.1	24.6	0.370
	(17.5–29.2)	(9.6)	(17.8–31.5)	(9.2)	
(6) Masticatory function *	107.0	108.5	122.5	124.5	0.028 *
	(72.5–134.3)	(46.4)	(75.5–171.8)	(56.8)	
(7) Swallowing function	1	2.8	0	2.2	0.329
	(0–3)	(5.1)	(0–1)	(5.4)	
Number of hypofunctional oral examination items	3.5	3.7	4	3.5	0.319
	(3–4)	(0.9)	(2–4)	(1.4)	

* *p* < 0.05; Wilcoxon signed rank test (1st vs. 2nd). Hypo group, patients with oral hypofunction; ODK, oral diadochokinesis; Q, quartile; SD, standard deviation.

**Table 3 ijerph-19-02154-t003:** Temporal changes in oral function between 1st and 2nd examination in the Pre-hypo group.

Pre-Hypo Group (N = 26)	1st Examination		2nd Examination		
Seven Oral Examination Items for Oral Hypofunction	Median (Q1–Q3)	Mean (SD)	Median (Q1–Q3)	Mean (SD)	*p*-Value
(1) Oral hygiene *	22.2 (9.7–29.2)	20.9 (15.9)	27.8 (16.7–45.8)	32.9 (23.5)	0.043 *
(2) Oral wetness *	30.1 (28.0–31.0)	30.0 (3.1)	28.0 (26.9–30.1)	28.0 (2.4)	0.004 *
(3) Occlusal force	661.5 (527.8–947.5)	715.8 (324.8)	629.0 (404.5–952.0)	696.3 (397.5)	1.000
(4) Tongue-lip motor function					
ODK /pa/	6.2 (6.0–7.0)	6.4 (0.7)	6.0 (5.6–6.7)	6.1 (0.7)	0.112
ODK /ta/	6.2 (6.0–6.7)	6.3 (0.7)	6.2 (5.6–6.7)	6.2 (0.8)	0.051
ODK /ka/	6.0 (5.6–6.4)	5.9 (0.7)	5.8 (5.0–6.3)	5.7 (0.8)	0.084
(5) Tongue pressure	27.2 (21.9–32.2)	27.9 (6.9)	27.8 (23.6–31.2)	27.9 (6.0)	0.858
(6) Masticatory function	125.0 (115.8–155.8)	137.9 (36.8)	149.0 (101.5–187.8)	152.5 (64.4)	0.261
(7) Swallowing function	0 (0–0)	0.5 (1.2)	0 (0–0)	0.4 (0.8)	0.828
Number of hypofunctional oral examination items *	2 (1–2)	1.5 (0.7)	2 (2–3)	2.2 (1.2)	0.004 *

* *p* < 0.05; Wilcoxon signed rank test (1st vs. 2nd). Hypo group, patients with oral hypofunction; ODK, oral diadochokinesis; Q, quartile; SD, standard deviation.

**Table 4 ijerph-19-02154-t004:** Interaction effect in temporal changes of oral function between 1st and 2nd examination in the Hypo and Pre-hypo groups.

Seven Oral Examination Items for Oral Hypofunction	Hypo Group	Pre-Hypo Group	F-Value	*p*-Value
(1) Oral hygiene		Declined	3.415	0.191
(2) Oral wetness		Declined	1.583	0.213
(3) Occlusal force			1.759	0.189
(4) Tongue-lip motor function				
ODK /pa/ *	Improved		11.438	0.001 *
ODK /ta/ *	Improved		6.291	0.015 *
ODK /ka/			3.227	0.077
(5) Tongue pressure			0.154	0.696
(6) Masticatory function	Improved		0.010	0.922
(7) Swallowing function			0.238	0.627
Number of hypofunctional oral examination items *		Declined	8.671	0.004 *

* *p* < 0.05; two-way repeated measures ANOVA. Hypo group, patients with oral hypofunction; Pre-hypo group, patients without oral hypofunction; ODK, oral diadochokinesis.

**Table 5 ijerph-19-02154-t005:** Prevalence rate of oral hypofunction in the first and second examination.

Group	Hypo Group at 2nd Exam N (%)	Pre-Hypo Group at 2nd Exam N (%)	Sum N (%)
Hypo group at 1st exam	31 (73.8)	11 (26.2)	42 (100)
Pre-hypo group at 1st exam	12 (46.2)	14 (53.8)	26 (100)
Sum	43 (100)	25 (100)	68

Hypo group, patients with oral hypofunction; pre-hypo group, patients without oral hypofunction.

**Table 6 ijerph-19-02154-t006:** Multiple regression analysis of the amount of improvement in the number of hypofunctional items for oral hypofunction between the 1st and 2nd examinations.

Explanatory Variables	B	SE	β	*p*-Value	95% CI	VIF
Age	0.035	0.022	0.201	0.117	−0.009 to 0.078	1.185
Sex	−0.009	0.340	−0.003	0.979	−0.689 to 0.671	1.118
Measurement period	−0.118	0.085	−0.181	0.170	−0.287 to 0.052	1.258
Number of visits	0.084	0.055	0.315	0.133	−0.026 to 0.194	3.189
Dental treatment	−0.203	0.309	−0.131	0.514	−0.822 to 0.415	2.942
CCI	0.068	0.203	0.041	0.738	−0.337 to 0.474	1.106
Oral health management for oral hypofunction *	−1.104	0.342	−0.388	0.002 *	−1.788 to −0.420	1.077

* *p* < 0.05; R^2^ = 0.193. The objective variable was the amount of improvement in the number of hypofunctional items for oral hypofunction from the 1st to the 2nd examination. CCI, Charlson comorbidity index; SE, standard error; CI, confidence interval; VIF, variance inflation factor. Sex (0 = men, 1 = women); Dental treatment (0 = maintenance only, 1 = minor change, 2 = major change); Oral health management for oral hypofunction (0 = Pre-hypo group, 1 = Hypo group).

## Data Availability

Data is not available due to ethical problems.

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
