# Peer review of "Regular Oral Health Management Improved Oral Function of Outpatients with Oral Hypofunction in Dental Hospital: A Longitudinal Study"

_ijerph, 2022, doi:10.3390/ijerph19042154_

Round 1
Reviewer 1 Report
The concept of oral hypofunction, a comprehensive examination of oral function in the older adults, is revolutionary. However, the evidence for oral hypofunction is lacking, and the methods for dealing with it have not been established. This paper is an important study that shows one way to deal with oral hypofunction. This paper is already of a quality that can be published in IJERPH, but in order to further improve the quality of the paper, please consider the following arguments from me.
Major points
- Even though the study design is different from your previous study(Hatanaka Y, Furuya J, Sato Y, Uchida Y, Osawa T, Shichita T, Suzuki H, Minakuchi S. Impact of oral health guidance on the tongue-lip motor function of outpatients at a dental hospital. Gerodontology. 2021 Oct 24. doi: 10.1111/ger.12599. Epub ahead of print. PMID: 34689371.), it gives the illusion that the target audience and the method of instruction using the leaflet are identical. Please add the uniqueness of your study that differs from previous studies.
- I feel uncomfortable with the mixture of the non-parametric tests, Mann-Whitney U test and Wilcoxon's signed-rank test, and the parametric tests, two-factor repeated measures ANOVA and linear multiple regression analysis. If parametric tests are to be used in the final analysis, why not unify them with parametric tests to imply that the measurement results are normally distributed?
- The authors consider that the key to the improvement in oral functions is the improvement in tongue and lip motor function. This is a very interesting observation. However, I think it is a limitation that the authors did not evaluate how much the frequency of conversation opportunities, the instructional method of the leaflet used in this study, changed before and after the intervention.
Minor points
- Please standardize the terminology in the manuscript. For example, men/women and male/female, older people and older adults, remaining teeth and present teeth, etc.
- The reference cited in the oral moisture meter does not seem to describe the method used by the authors. Please check the references again to make sure they are correct.
- Please check again whether the results of the chi-square test include results that are appropriate to be examined by Fisher's exact test.
- Isn't the serial number of "3.1. Comparison of the first and second times in the Hypo and Pre-hypo groups" in the Results a mistake for 3.2?
- The sex in Table 6 should describe the assignment of the dummy variable.
Author Response
Point-by-point response to Reviewer 1
Comment #1
The concept of oral hypofunction, a comprehensive examination of oral function in the older adults, is revolutionary. However, the evidence for oral hypofunction is lacking, and the methods for dealing with it have not been established. This paper is an important study that shows one way to deal with oral hypofunction. This paper is already of a quality that can be published in IJERPH, but in order to further improve the quality of the paper, please consider the following arguments from me.
Response #1
Thank you for your kind and thoughtful review of our manuscript. We appreciate your insightful comments, and have revised the manuscript according to your suggestions and comments. We have considered your opinion and worked hard to improve the manuscript for the readers of IJERPH.
Comment #2
Major points
Even though the study design is different from your previous study (Hatanaka Y, Furuya J, Sato Y, Uchida Y, Osawa T, Shichita T, Suzuki H, Minakuchi S. Impact of oral health guidance on the tongue-lip motor function of outpatients at a dental hospital. Gerodontology. 2021 Oct 24. doi: 10.1111/ger.12599. Epub ahead of print. PMID: 34689371.), it gives the illusion that the target audience and the method of instruction using the leaflet are identical. Please add the uniqueness of your study that differs from previous studies.
Response #2
Thank you for your kind and thoughtful comment. We appreciate the Reviewer taking the time to read our previous research in Gerodontology. As mentioned, we should emphasize the strength and novelty of our present research. In the present research, we provided regular comprehensive oral health management including dental treatment. In addition, we observed the temporal change in oral function in those without oral hypofunction, suggesting that their oral function tends to decline due to the absence of oral health instruction. We agree with your comment and have revised the Introduction and Discussion sections to emphasize it as follows:
L59
“Shirobe et al. [13] found that an oral frailty prevention program (comprehensive oral function training) at the dental office improved the oral function of community-dwelling older people. We also clarified the effect of oral health instruction on oral diadochokinesis using a leaflet on dental care in older dental outpatients with de-creased tongue-lip motor function, an examination item for oral hypofunction [9].
However, the effectiveness of regular and comprehensive oral health management for oral hypofunction, including dental treatment and systemic diseases, remains unknown. Dental treatment such as prosthetic treatment could improve the criteria for oral hypofunction, such as the occlusal force and masticatory function. In addition, systemic diseases could affect almost all the criteria for oral hypofunction. Therefore, this longitudinal study aimed to investigate the impact of regular comprehensive oral health management for oral hypofunction in older dental outpatients in daily dental care. “
L263
“To manage oral hypofunction, it is necessary that patients accept and actively work on improving their declining oral function. It is extremely important to maintain the patients' awareness and motivation for oral function. The results of this study revealed that several oral functions can be improved or maintained by providing regular comprehensive oral health instructions using a leaflet in patients with oral hypofunction in a dental outpatient clinic. The oral health instruction would help motivate patients to improve their oral function in their daily life. In particular, tongue-lip motor function, for which patients can easily perform daily training, was found to improve even within a short period of time. In addition, it could be considered that the improvement in tongue-lip motor function led to improvement in the masticatory function. Conversely, patients who did not receive regular oral health instruction were more likely to have a decline in oral function over time. This tendency was particularly pronounced in the oral environment, including oral hygiene and wetness. Oral function can be measured using the seven criteria for oral hypofunction; however, oral function itself is a comprehensive function. The seven oral examination items are interrelated; therefore, it is important to provide comprehensive oral health instruction for oral function. The importance of oral health instruction for oral function in dental outpatient clinics as well as conventional dental health guidance for oral hygiene was clarified. Oral hypofunction, which is considered a condition in which oral frailty has progressed, may contribute to the prevention of physical and social frailty through the diversity of food intake and malnutrition. Therefore, regular oral health management for oral hypofunction in dental outpatient clinics will be very important in the future super-aging society, and this study is the first to clarify its effect. ”
L294
“Therefore, motivating patients to maintain and improve oral function through self-training and dental treatment, as well as providing comprehensive oral health in-structions including lifestyle guidance, and nutritional counseling, are recommended as methods to manage oral hypofunction [25]. ”
L344
“Previous studies have shown that oral dryness worsens tongue coating [36], suggesting that oral hygiene and wetness in oral hypofunction were related. “
L347
“Oral hypofunction was diagnosed when three or more of the seven items in each examination were hypofunctional. Therefore, the number of items that fall under the declined category in each examination is important, as it may show the severity of oral hypofunction. In this study, the number of hypofunctional items changed differently over time in both groups, and multiple analyses revealed that the number of hypofunctional items tended to decrease in the Hypo group, suggesting that regular oral health management for oral hypofunction might be effective for older dental outpatients. Approximately a quarter of the Hypo group recovered from oral hypofunction. In contrast, the number of hypofunctional items was significantly increased in the Pre-hypo group, and half of the Pre-hypo group was newly diagnosed with oral hypofunction at the second examination. Therefore, it is suggested that the absence of oral health management could lead to a decline in oral function due to a decrease in the motivation of older outpatients. Our previous study [9] revealed the importance of reconsidering the cut-off value of the seven items and the criterion for three out of seven items necessary for diagnosis. In addition, our previous study revealed several oral examination items for oral hypofunction indicated a relationship between age and sex. Although it is still debatable whether the cut-off values of seven items were validated for diagnosis, seven oral examination items were needed to understand how their oral function declined and how to improve them individually. Thus, the present study suggested the necessity of regular oral health management for older patients, including those without an oral hypofunction diagnosis as well as those with oral hypofunction.“
Comment #3
I feel uncomfortable with the mixture of the non-parametric tests, Mann-Whitney U test and Wilcoxon's signed-rank test, and the parametric tests, two-factor repeated measures ANOVA and linear multiple regression analysis. If parametric tests are to be used in the final analysis, why not unify them with parametric tests to imply that the measurement results are normally distributed?
Response #3
Thank you for your valuable comment regarding the statistical methods. Some of our result include informally distributed data; therefore, we decided to use non-parametric tests for univariate analysis. However, to improve the number of hypofunctional items, and the difference in the values in the oral exam items from the first examination to the second examination were considered to be normally distributed. Therefore, we used the parametric tests, ANOVA and multiple analysis, to clarify the difference in the temporal change in the Hypo and Pre-hypo group and the factors associated with the improvement in the number of hypofunctional items between the two examinations. We also referred to the statistical method used in the study by Matsubara C, Shirobe M, Furuya J, Watanabe Y, Motokawa K, Edahiro A, Ohara Y, Awata S, Kim H, Fujiwara Y, Obuchi S, Hirano H, Minakuchi S. Effect of oral health intervention on cognitive decline in community-dwelling older adults: A randomized controlled trial. Arch Gerontol Geriatr. 2021 Jan-Feb;92:104267. doi: 10.1016/j.archger.2020.104267. Epub 2020 Sep 28. PMID: 33035763.
Comment #4
The authors consider that the key to the improvement in oral functions is the improvement in tongue and lip motor function. This is a very interesting observation. However, I think it is a limitation that the authors did not evaluate how much the frequency of conversation opportunities, the instructional method of the leaflet used in this study, changed before and after the intervention.
Response #4
Thank you for your valuable advice. We agree with your comment that it would be an important limitation of the present study. We have added this information in the study limitation section as follows:
“Additionally, we did not evaluate the change in the frequency of conversation opportunities and other related factors which were mentioned in the leaflet used in this study and would affect the result of this study.”
Comment #5
Minor points
- Please standardize the terminology in the manuscript. For example, men/women and male/female, older people and older adults, remaining teeth and present teeth, etc.
Response #5
Thank you for pointing out the inconsistency in our manuscript. We have standardized the terminology as follows:
Men/women; older individuals or older patients; teeth present
Comment #6
- The reference cited in the oral moisture meter does not seem to describe the method used by the authors. Please check the references again to make sure they are correct.
Response #6
Thank you for your constructive comment. We have added the correct reference according to your comment.
Comment #7
- Please check again whether the results of the chi-square test include results that are appropriate to be examined by Fisher's exact test.
Response #7
Thank you for pointing out this error. We have revised Table 1, and added the correct statistical method according to your comment.
Comment #8
- Isn't the serial number of "3.1. Comparison of the first and second times in the Hypo and Pre-hypo groups" in the Results a mistake for 3.2?
- The sex in Table 6 should describe the assignment of the dummy variable.
Response #8
Thank you for pointing out these errors in our manuscript. We have revised the manuscript according to your comment.

Reviewer 2 Report
Congratulations to Authors for this very interesting manuscript, well written and with a good metodological design. The topic shows a good appropriateness for IJERPH. The Results, Discussion and Conclusions are clear. The limits of the study have been showed clearly. I would suggest to Authors to continue this study considering the oral-chewing function and cognitive aspects of elderly patients. About references they should check the style of all (year in bold and n.10 abbreviation of journal). In Abstract: row 16 > Define Acronym ODK. Pag.2-row 57> citation [13] after et al.
Author Response
Point to point response to Reviewer 2
Comment #1
Congratulations to Authors for this very interesting manuscript, well written and with a good methodological design. The topic shows a good appropriateness for IJERPH. The Results, Discussion and Conclusions are clear. The limits of the study have been showed clearly. I would suggest to Authors to continue this study considering the oral-chewing function and cognitive aspects of elderly patients. About references they should check the style of all (year in bold and n.10 abbreviation of journal). In Abstract: row 16 > Define Acronym ODK. Pag.2-row 57> citation [13] after et al.
Response #1
Thank you for your kind review of our manuscript. We appreciate your thoughtful comments and have revised the manuscript according to your comments. Unfortunately, we could not consider the cognitive function of the older patients in the present study. We took dementia into account while considering CCI, and almost all participants had good cognitive function. We agree with your opinion and we shall consider them in the future.

Reviewer 3 Report
Dear Authors,
the article entitled "Regular oral health management improved oral function of outpatients with oral hypofunction in dental hospital: a longitudinal study" aims to investigate oral declining in super-aging societies. The manuscript evidences the importance of oral frailty and analyzes the progressive decline in oral function, ranging from trivial decline to dysphagia.
The study participants comprised 105 patients who visited the outpatient clinic of geriatric dentistry and who underwent oral examinations for oral hypofunction for the first time.
This study is based on seven oral examination criteria: oral hygiene, oral wetness, occlusal force, tongue-lip motor function, tongue pressure, masticatory function, and swallowing function.
The participants, according to the number of declined oral examinations, were included in two groups: the oral hypofunction group (Hypo group, >3 criteria ) and pre-oral hypofunction group (Pre-hypo group, ≤2 criteria).
In conclusion: after a detailed analysis of data, geriatric patients have had a significant increase in oral health following methodic oral health management for oral hypofunction, Tongue-lip motor function, and masticatory function.
To improve your manuscript, I suggest you implement the following points, in particular in:
Introduction
Requires a better definition of the aim of the project.
Materials and methods
Row 80 “The results of this study are shown in Fig. 1”
Fig 1 is not present in the article.
In my opinion, it could be better to evidence which are the seven oral examination criteria in a table individuated for each group like this
|
Examination criteria: |
Hypo group (42) |
Pre-hypo group(26) |
Sum (68) |
|
1) oral hygiene |
x |
value |
|
|
2) oral wetness |
y |
value |
|
|
3) occlusal force |
z |
value |
|
|
4) tongue-lip motor function |
k |
value |
|
|
5) tongue pressure |
q |
value |
|
|
6) masticatory function |
w |
value |
|
|
7) swallowing function |
j |
value |
|
In discussion
In my opinion, the Authors should explain the importance of the seven criteria, the motivation to have chosen that, and the relationships among those criteria and the older people.
Best Regards
Author Response
Point to point response to Reviewer 3
Comment #1
Dear Authors,
The article entitled "Regular oral health management improved oral function of outpatients with oral hypofunction in dental hospital: a longitudinal study" aims to investigate oral declining in super-aging societies. The manuscript evidences the importance of oral frailty and analyzes the progressive decline in oral function, ranging from trivial decline to dysphagia.
The study participants comprised 105 patients who visited the outpatient clinic of geriatric dentistry and who underwent oral examinations for oral hypofunction for the first time.
This study is based on seven oral examination criteria: oral hygiene, oral wetness, occlusal force, tongue-lip motor function, tongue pressure, masticatory function, and swallowing function.
The participants, according to the number of declined oral examinations, were included in two groups: the oral hypofunction group (Hypo group, >3 criteria ) and pre-oral hypofunction group (Pre-hypo group, ≤2 criteria).
In conclusion: after a detailed analysis of data, geriatric patients have had a significant increase in oral health following methodic oral health management for oral hypofunction, Tongue-lip motor function, and masticatory function.
Response #1
Thank you for your kind and thoughtful review of our manuscript. As mentioned, we focused on oral hypofunction diagnosed using seven oral examination items; therefore, dentists could consider the decline in oral function in older individuals in their daily dental care. We greatly appreciate your constructive comments and have revised the manuscript according to your comments.
Comment #2
To improve your manuscript, I suggest you implement the following points, in particular in:
Introduction
Requires a better definition of the aim of the project.
Response #2
Thank you for your constructive comment. We agree with your comment. Therefore, we have revised the Introduction section to ensure that the readers can understand the purpose of this study.
L59
“Shirobe et al. [13] found that an oral frailty prevention program (comprehensive oral function training) at the dental office improved the oral function of community-dwelling older people. We also clarified the effect of oral health instruction on oral diadochokinesis using a leaflet on dental care in older dental outpatients with de-creased tongue-lip motor function, an examination item for oral hypofunction [9].
However, the effectiveness of regular and comprehensive oral health management for oral hypofunction, including dental treatment and systemic diseases, remains unknown. Dental treatment such as prosthetic treatment could improve the criteria for oral hypofunction, such as the occlusal force and masticatory function. In addition, systemic diseases could affect almost all the criteria for oral hypofunction. Therefore, this longitudinal study aimed to investigate the impact of regular comprehensive oral health management for oral hypofunction in older dental outpatients in daily dental care. “
Comment #3
Materials and methods
Row 80 “The results of this study are shown in Fig. 1”
Fig 1 is not present in the article.
Response #3
We have revised the manuscript as follows: “The results of this study are shown in Fig. 1” to “The characteristics of the participants of this study are shown in Table 1.”
Comment #4
In my opinion, it could be better to evidence which are the seven oral examination criteria in a table individuated for each group like this…
Response #4
We agree with your comment. We have revised Tables 2 and 3 according to your comment. We have numbered each oral examination items for oral hypofunction as you mentioned to ensure that the readers could easily understand the results. However, the primary objective of this study is to compare the second examination with the first examination in the Hypo group and Pre-hypo group; therefore, we have shown the results of the Hypo group and Pre-hypo group, respectively.
Comment #5
In discussion
In my opinion, the Authors should explain the importance of the seven criteria, the motivation to have chosen that, and the relationships among those criteria and the older people.
Best Regards
Response #5
We agree with your comment. We have revised the Discussion section, focusing particularly on the first paragraph as follows:
L263
“To manage oral hypofunction, it is necessary that patients accept and actively work on improving their declining oral function. It is extremely important to maintain the patients' awareness and motivation for oral function. The results of this study revealed that several oral functions can be improved or maintained by providing regular comprehensive oral health instructions using a leaflet in patients with oral hypofunction in a dental outpatient clinic. The oral health instruction would help motivate patients to improve their oral function in their daily life. In particular, tongue-lip motor function, for which patients can easily perform daily training, was found to improve even within a short period of time. In addition, it could be considered that the improvement in tongue-lip motor function led to improvement in the masticatory function. Conversely, patients who did not receive regular oral health instruction were more likely to have a decline in oral function over time. This tendency was particularly pronounced in the oral environment, including oral hygiene and wetness. Oral function can be measured using the seven criteria for oral hypofunction; however, oral function itself is a comprehensive function. The seven oral examination items are interrelated; therefore, it is important to provide comprehensive oral health instruction for oral function. The importance of oral health instruction for oral function in dental outpatient clinics as well as conventional dental health guidance for oral hygiene was clarified. Oral hypofunction, which is considered a condition in which oral frailty has progressed, may contribute to the prevention of physical and social frailty through the diversity of food intake and malnutrition. Therefore, regular oral health management for oral hypofunction in dental outpatient clinics will be very important in the future super-aging society, and this study is the first to clarify its effect. ”
L294
“Therefore, motivating patients to maintain and improve oral function through self-training and dental treatment, as well as providing comprehensive oral health in-structions including lifestyle guidance, and nutritional counseling, are recommended as methods to manage oral hypofunction [25]. ”
L344
“Previous studies have shown that oral dryness worsens tongue coating [36], suggesting that oral hygiene and wetness in oral hypofunction were related. “
L347
“Oral hypofunction was diagnosed when three or more of the seven items in each examination were hypofunctional. Therefore, the number of items that fall under the declined category in each examination is important, as it may show the severity of oral hypofunction. In this study, the number of hypofunctional items changed differently over time in both groups, and multiple analyses revealed that the number of hypofunctional items tended to decrease in the Hypo group, suggesting that regular oral health management for oral hypofunction might be effective for older dental outpatients. Approximately a quarter of the Hypo group recovered from oral hypofunction. In contrast, the number of hypofunctional items was significantly increased in the Pre-hypo group, and half of the Pre-hypo group was newly diagnosed with oral hypofunction at the second examination. Therefore, it is suggested that the absence of oral health management could lead to a decline in oral function due to a decrease in the motivation of older outpatients. Our previous study [9] revealed the importance of reconsidering the cut-off value of the seven items and the criterion for three out of seven items necessary for diagnosis. In addition, our previous study revealed several oral examination items for oral hypofunction indicated a relationship between age and sex. Although it is still debatable whether the cut-off values of seven items were validated for diagnosis, seven oral examination items were needed to understand how their oral function declined and how to improve them individually. Thus, the present study suggested the necessity of regular oral health management for older patients, including those without an oral hypofunction diagnosis as well as those with oral hypofunction.“

Round 2
Reviewer 3 Report
Dear authors,
Thank you for your prompt response and revision. The manuscript results are now improved and clearer. However, a few corrections are still needed:
Introduction:
at line 62 the concept is not clear, is it part of the aim or of the material and methods?
line 57: Please do not use colloquialism such as "To our knowledge"
The rest of the manuscript is well-written and linear.
Best regards
Author Response
Please see the attachment file of our point to point response. We put the content here as follows;
Point to point response to Reviewer 3
Comment #1
Dear authors,
Thank you for your prompt response and revision. The manuscript results are now improved and clearer. However, a few corrections are still needed:
Introduction:
at line 62 the concept is not clear, is it part of the aim or of the material and methods?
line 57: Please do not use colloquialism such as "To our knowledge"
The rest of the manuscript is well-written and linear.
Best regards
Response #1
Thank you for kind review. We appreciate your comments and revised the manuscript according to your comment as follows;
L57 “To our knowledge” was deleted.
L62 We revised the manuscript and added the detail explanation of our concept so that readers could understand the concept easily. The revised sentences were highlighted in yellow. Thank you in advance.
“We also clarified the effect of oral health instruction on oral diadochokinesis in our previous study [9], indicating that decreased tongue-lip motor function in older dental outpatients could be improved by oral health instruction.
However, the effectiveness of regular and comprehensive oral health management for oral hypofunction, including the influence of dental treatment and presence of systemic diseases on oral function, remains unknown.Dental treatment such as prosthetic treatment could improve the criteria for oral hypofunction, such as the occlusal force and masticatory function. In addition, systemic diseases could affect almost all the criteria for oral hypofunction. Although these factors would affect the regular oral health management of oral hypofunction in daily dental care, previous researches did not consider them in detail. Therefore, this longitudinal study aimed to investigate the impact of regular comprehensive oral health management for oral hypofunction in older dental outpatients in daily dental care.”
